# Multilingual Offline Signature Verification Based on Improved Inverse Discriminator Network

**Nurbiya Xamxidin [1], Mahpirat [2], Zhixi Yao [1], Alimjan Aysa [1] and Kurban Ubul [1,3,\*]**

[1] School of Information Science and Engineering, Xinjiang University, Urumqi 830046, China; nurbiyaaa@stu.xju.edu.cn (N.X.); yzhx0511@stu.xju.edu.cn (Z.Y.); alim@xju.edu.cn (A.A.)
[2] Educational Administration Department, Xinjiang University, Urumqi 830046, China; xmahpu@xju.edu.cn
[3] Xinjiang Multilingual Information Technology Key Laboratory, Urumqi 830046, China
[\*] Correspondence: kurbanu@xju.edu.cn

**Abstract:** To further improve the accuracy of multilingual off-line handwritten signature verification, this paper studies the off-line handwritten signature verification of monolingual and multilingual mixture and proposes an improved verification network (IDN), which adopts user-independent (WI) handwritten signature verification, to determine the true signature or false signature. The IDN model contains four neural network streams with shared weights, of which two receiving the original signature images are the discriminative streams, and the other two streams are the reverse stream of the gray inversion image. The enhanced spatial attention models connect the discriminative streams and reverse flow to realize message propagation. The IDN model uses the channel attention mechanism (SE) and the improved spatial attention module (ESA) to propose the effective feature information of signature verification. Since there is no suitable multilingual signature data set, this paper collects two language data sets (Chinese and Uyghur), including 100,000 signatures of 200 people. Our method is tested on the self-built data set and the public data sets of Bengali (BHsig-B) and Hindi (BHsig-H). The method proposed in this paper has the highest discrimination rate of FRR of 10.5%, FAR of 2.06%, and ACC of 96.33% for the mixture of two languages.

**Keywords:** multilingual offline signature; IDN network; channel attention mechanism (SE); improved spatial attention mechanism (ESA)

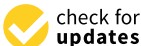



## 1. Introduction

Biometrics is an innate characteristic of each person or related to personal habits that have been formed over time, and it is a highly secure and difficult-to-imitate pattern recognition method. As a legally recognized biometric, handwritten signatures can be divided into handwritten signature identification and handwritten signature verification according to different research contents and research methods [1]. The overall experimental process of the two is very similar. The former is mainly used to distinguish which user all signature images belong to, while the latter is mainly used to distinguish whether all signature images in the same category are the real signature of that person. Handwritten signatures can be collected online or offline in different ways [2]. The offline handwritten signature verification system can be divided into writer dependent (WD) and writer independent (WI) [3]. This paper studies the handwritten signature verification independent of the writer (WI), distinguishing between genuine and false signatures, as shown in Figure 1. So far, great achievements have been made in the research of handwritten signatures of Chinese, Latin, and other mainstream characters in the world. At present, there are few studies on handwritten signatures in Chinese minority languages, and there is no suitable multilingual signature dataset. Secondly, there are also sparse signature images, because the strokes of signatures are often very thin, and most of the signature images are empty, it is difficult to extract effective features. and finally, most people's signature style has a

certain arbitrariness, distinguishing between true and false signatures is more male, and the signature identification rate is low.

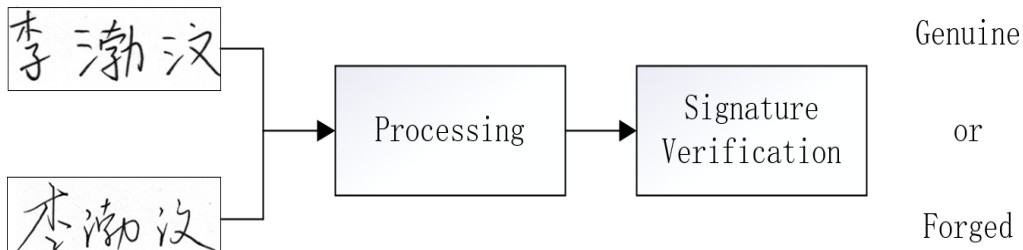

**Figure 1.** The overall framework of the verification system. (Example of Chinese signature in the figure: Li Bowen).

In this paper, we propose an improved inverse discriminative network (IDN) for experimentation with a user-independent (WI) handwriting signature verification method.

The IDN model consists of four weight-sharing streams, two of which are discriminative streams and two are reverse streams, of which two discriminative streams receive the reference signature and test signature, which are used as input during the experiment and extract useful features of the signature through four cascading convolutional modules, and the discriminative streams also receive a grayscale of the reference signature and the test signature. The two reverse streams are connected by an improved spatial attention mechanism that propagates information at different scales, reinforcing valid signature information. The features from the discriminating and reverse streams are merged into three different feature maps by convolutional modules and then sent to the fully connected layer for decision making. The IDN model introduces two mechanisms to solve the problem of sparse information of signatures: the first is the channel attention mechanism SE module, through which the network can learn to use global information to selectively emphasize the information characteristics of the signature that are valid, and suppress less useful features; the second is the improved spatial attention mechanism ESA module, which propagates messages between reverse and discriminating flows through attention modules of different feature scales. The purpose of the attention mechanism is to enhance model learning and extract important features of signature verification. Our contributions are as follows:

- We build a multi-language offline handwritten signature database, use PS to segment the signature, and perform preprocessing operations.
- For the IDN model, the channel attention machine and the improved spatial attention mechanism are used to further improve the handwritten signature identification.
- The method proposed in this paper can effectively improve the handwritten signature verification rate of single and mixed languages, and pave the way for multilingual signatures such as Kazakh and Kirgiz.

## 2. Related Work

Abdul-Haleem et al. [4] define a new promising approach in handwritten signature verification systems that extracts signature features using the Haar wavelet technique and a set of local ridges features. Batool et al. [5] proposed a signature identification technique based on multi-level feature fusion and optimal feature selection. From the preprocessed feature samples, 22 grayscale occurrence matrix (GLCM) features and 8 geometric features are calculated, and the classifier uses SVM. In the data set, MCYT, CEDAR, and GPDS are used, among which: The dataset MCYT had a FAR of 2.66%, FRR of 2.00%; the dataset CEDAR had a FRR of 3.75%, FAR of 3.34%; and the dataset GPDS had a FRR of 9.69%, FAR of 10.3%. M. Ajij et al. [6] used a signature verification feature set based on the operating quasi linearity of boundary pixels for classification using a support vector machine (SVM) and showed the corresponding results on standard signature datasets such as Cedar and GPDS-100. FAR = 15.04%, FRR = 7.85%, and AER = 12.42%. Kumar A et al. [7], using the

pixels located on the elliptic curve path to extract the features of the authentic and fake signature images, established a multi-classifier system and used two classifiers, namely the polynomial kernel support vector machine and the quadratic support vector machine classifier. To generate MCS, SVMP and SVMQ classifiers are created. Arab et al. [8] proposed a new texture feature method local difference feature, LDF. LDF calculated the difference between the center pixel and the 8 fields on the specific field radius, The classifier used the SVM, with two public datasets, GPDS and CEDAR, with their AER of 6.10% and 6.10%, respectively. Liu, L. et al. [9] proposed a novel framework for metric learning; in experiments on the benchmark datasets CEDAR and GPDS, the EER was 4.55% and 8.89%, respectively. Natarajan et al. [10] designed a mutual signature dense Network (MSDN) to extract features. In experiments on the public datasets CEDAR and GPDS, the proposed approach achieved the latest performance of 6.74% and 8.24% EER, respectively, in the WI scenario, and 1.67% and 1.65% EER, respectively, in the WD scenario. Kennedy Gyimah et al. [11] proposed an improved feature extraction vector and offline signature verification system by combining the GLCM feature and image region feature. GPDS was used in the data set, and the discrimination rate was FAR = 2.5% and FRR = 14%. Chinmay, L. et al. [12] removed noise through a Gaussian filter and perform Gaussian difference processing on the image. The GLCM is used to extract the features of the denoised image, and then the extracted features are reduced by Principal Component Analysis (PCA) and Kernel Principal Component Analysis (KPCA). The data set uses Kaggle, the accuracy rate of the KNN classifier is 82%, and the accuracy rate of random forest is 81.66%. Masoudnia S et al. [13] conducted two sets of experiments on two different protocols of OSV with different loss functions of CNN, namely writer-dependent and writer-independent three signature datasets: GPDS synthesis, MCYT, and UT-SIG. Zhang, S.J. et al. [14] extracted the BoVW of the feature and selection algorithm MRMR for the Verification of Uyghur and CEDAR dataset identification of Uyghur and English signatures. To improve the correlation between feature vectors and categories, the maximum correlation and minimum redundancy algorithms are used to perform feature selection on visual word features.

However, the intra-class differences between real and forged signatures due to the sparsity of the effective feature information of handwritten signature images and the randomness of most people's signature styles remain an unsolvable problem in offline handwritten signature recognition and verification. In addition, there are no suitable public offline handwritten signature datasets for Mandarin Chinese and Chinese minority languages, which makes the study of offline handwritten signatures in different textual forms relatively straightforward. To solve the above problems, this paper establishes an offline handwritten signature of Chinese and Uyghur data sets and proposes an improved offline handwritten signature identification method based on IDN.

## 3. Multilingual Signature Verification

### 3.1. Signature Datasets

Since there is no suitable multi-language offline handwritten signature data set, this paper collects signature data sets in three different languages, including genuine signatures and forged signatures. For the collected self-built signatures, firstly, volunteers write their signatures on A4 paper 20 times, and each real signature has a corresponding forged signature. Forged signatures are simply forged by different volunteers using their writing styles and habits. These collected signatures are first converted by devices such as scanners into digital signals that can be processed by computers. Each signature has 20 genuine handwritten signature image samples and 20 forged handwritten signature image samples. The self-built multilingual dataset contains 10,000 signature samples of 200 individuals. Some examples are shown in Figure 2.

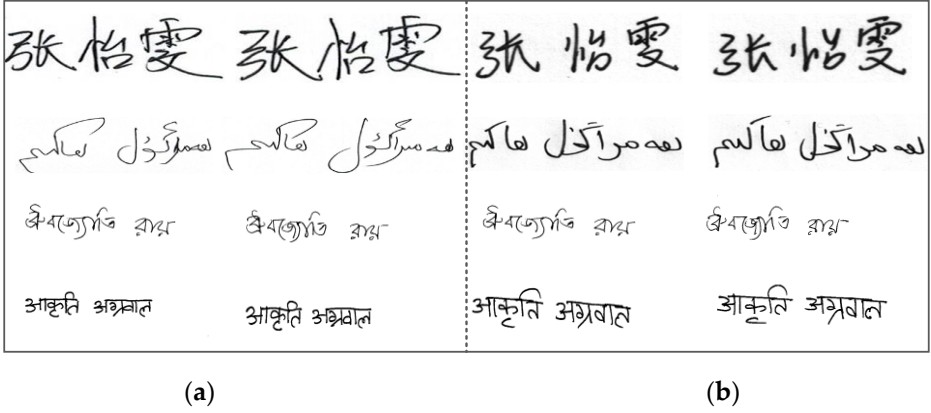

(**a**)          (**b**)

**Figure 2.** Some multi-language offline handwritten signatures: (**a**) an example of partial genuine signatures in Chinese, Uyghur, Bengali, and Indian languages; (**b**) an example of partial forged signatures in Chinese, Uyghur, Bengali, and Indian languages. (Example of Chinese signature in the figure: Zhang Yiwen).

### 3.2. Preprocessing

The task of preprocessing is to remove invalid information that is not discriminative during the identification process and to reduce the intra-class variability between images of genuine and fake signatures [15]. In practical applications, in the offline handwritten signature system, the signer signs on paper, and then all signature data is scanned into a static image, so the signature image may have changes in the background, pen thickness, scale, rotation, etc., To improve the validity of the model, the signature image model needs to be cropped and resized. These signature images are preprocessed by the Otsu algorithm and non-standard binarization. so that the background pixel value is 255 (white), and the signature stroke maintains the original gray value.

The datasets collected in this paper have several characteristics that make them unique and challenging. First of all, the self-built database is a large multi-language offline handwritten signature library, which we believe will help multi-language signature identification and other related research, and secondly, it has a large number of personal real signatures and corresponding forged signatures, which are collected manually in real life.

## 4. Proposed Method

### 4.1. Inverse Discriminative Networks

Signature strokes are the decisive feature that determines the identity of a signature. However, valid messages for signature verification in signature images are very sparse because the strokes of a signature are often very thin, while the signature image includes a large amount of invalid edge background. Therefore, the stroke of the signature is very important for offline handwritten signature research, because the signature stroke information is not used in a suitable model, and the effective signature features are not obtained, but the useless background information is obtained [16]. Therefore, to solve the problem that the model is mainly concerned with the stroke of the signature, this paper proposes an improved anti-authentication network (IDN). The basic idea of the IDN network is that when the grayscale values of the reference signature image and the test signature image are inverted, the model extracts the stroke characteristics of the signature and ignores the blank background of a large number of signature images and the model will make the same verification decision. Figure 3 is a reference handwritten signature and a test handwritten signature with black background, respectively, and two white, black background images are the inverse grayscale inversion images of the reference signature and the test signature respectively. For the four signature images to form three identification pairs, these three pairs of signature images as the input of the inverse identification model, through some column operations, produces a final output of three pairs of signature verification decisions, as shown in Figure 3: Decision 1, Decision 2, Decision 3. Since gray

values of the three e pairs of signature images are different, and the gray values are related to the stroke information of the signature, training the model with IDN will force the model to focus on the signature strokes without effectively suppressing the background information, and the identification rate is low. In order to improve this model structure, it can extract and learn complex feature information efficiently.

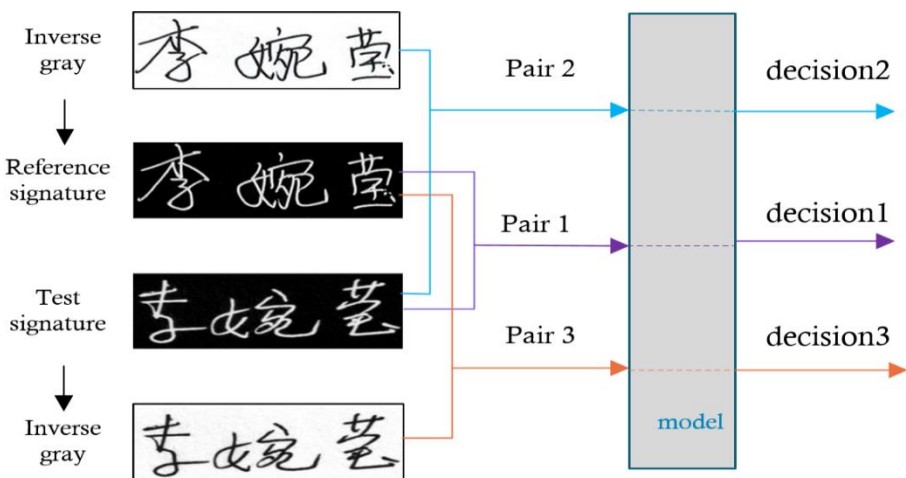

**Figure 3.** IDN network framework. (Example of Chinese signature in the figure: Li wanying).

If the model focuses on signature stroke information, it will make the same decision for 1 and 2, regardless of the invalid background of the signature image. Due to the difference between the three pairs of gray values, the common information is related to the stroke information of the signature, and training the model in this way will force the model to focus on the signature stroke and fail to effectively suppress the background information, and the identification rate is low, etc. This paper uses an improved model structure to extract and learn complex feature information efficiently.

### 4.2. Method Structure

The improved model based on IDN proposed in this paper is shown in Figure 4. The input to the model is the reference signature image, the test signature image, and the grayscale inversion of these two signature images, and the network consists of four weight-sharing streams. The two discriminative streams refer to the signature image and the test signature image as input, respectively, and extract signature features through a concatenated convolution module. Each convolution module consists of two convolutional layers activated by the ReLU function and a max-pooling layer. The two reverse streams take the inverse grayscale reference image and the test feature image as input, respectively, each reverse stream has the same structure as the discriminant stream, and there is an Enhanced Spatial attention mechanism (ESA) between the discriminant stream and the reverse stream to connect the convolution of the two streams modules; this paper also adopts the new unit structure of Squeeze-Excitation Block (SE), and uses the independence of the channel to readjust the characteristics of the channel dimension to obtain better results. Therefore, in order to further improve the representation ability of SE-block, an enhanced spatial attention mechanism (ESA) is introduced, which is used at the end of the residual block to force the features to focus more on the regions with important information. When the displayed features are aggregated together, we can get a more representative feature. The SE-block framework and ESA blocks are applied to build the final IDN network, each SE module contains 4 ESA blocks, and the ESA module contains a multi-layer convolutional neural network. As shown in Figure 4, each module consists of a forward process, which processes the feature outputs of the received convolutional network layers in the discriminative stream. The internal structure of the attention mechanism module will be introduced in detail in Section 4.4. Using three convolutional network layer

modules, the features of different streams are fused, in which reference signature features and test signature features, gray reference signature features and test signature features, the reference signature features and gray test signature features are extracted, respectively, and for three pairs of signatures, the image features are separately fused, and finally each fused feature is passed through a global average pool (GAP) layer, and the merged three features are input into three fully connected layers respectively, and the verification result is calculated.

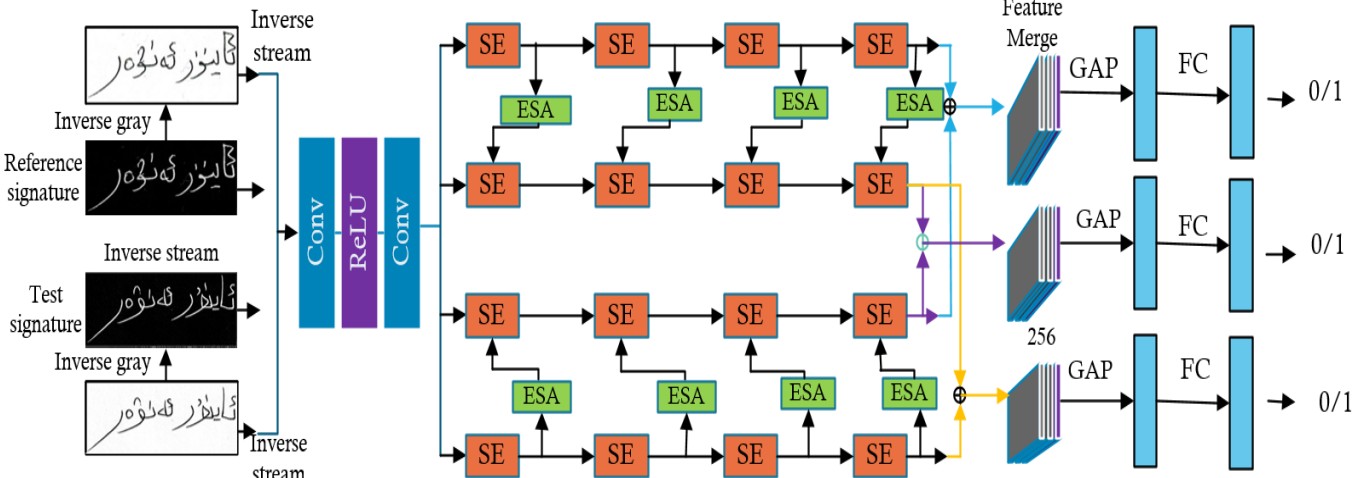

**Figure 4.** Proposed signature verification system.

Throughout the IDN architecture, discriminating flows and reverse flows are tightly linked by the ESA, and through these connections, the entire IDN model is trained in an end-to-end manner. The model uses two mechanisms to force the model to focus on the most efficient information features of the signature, SE-block, which enhances the learning of convolutional features by showing modeling channel correlation, enabling the network to increase its informative signatures on strokes. The second is ESA's improved spatial attention mechanism, which enables the model to extract important features of signature verification. As can be seen from Figure 5, some feature mappings of the output of the cascading convolutional module identify the stream.

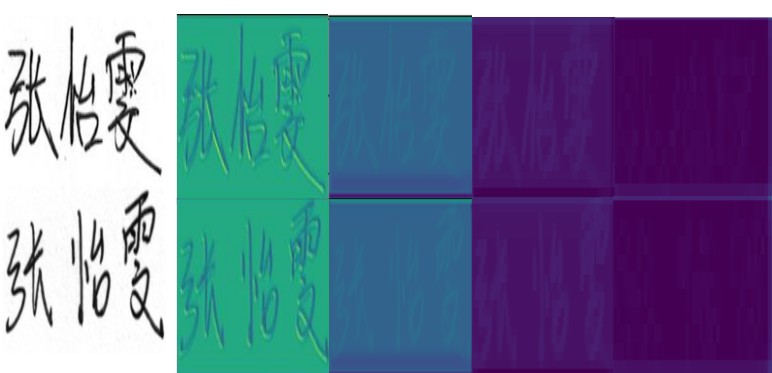

**Figure 5.** Diagram of the cascaded convolution module.

### 4.3. Channel Attention Mechanism

To maximize the effectiveness of the IDN framework in this paper, it is best to use it in combination with the attention mechanism, because we need to extract the valid strokes in the static signature image to obtain the highest signature discrimination rate [17]. This paper proposes a new unit structure SE-block, the SE-block which can be better modeled by taking advantage of the independence between feature channels of the convolution feature channel

dimension [18]. Therefore, this paper adopts a mechanism for feature readjustment, that is, it has the importance of each feature channel to sort, and the enhancement is important and the weakening is not important. Figure 6 shows the whole structure diagram of the SE module:

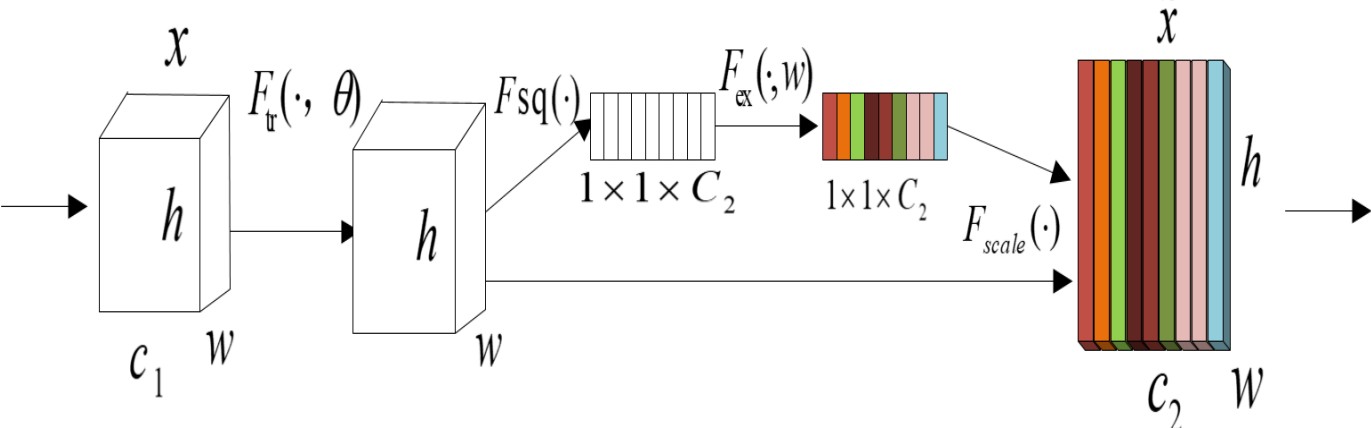

**Figure 6.** SE module structure diagram.

The above figure is a schematic diagram of the SE module, given an input x, the number of feature channels is $C_1$, and after a series 0 of convolutional and other general transformations, a feature with several feature channels is $C_2$, and the traditional CNN is different from the previously obtained features by recalibrating the previously obtained features through the following three operations [19]:

- Squeeze: compress H × W × C feature map 1 × 1 × C.
- Excitation: With the FC full connectivity layer, you can learn the importance of the feature channel. Different weights are assigned according to their importance.
- Reweight: this operation recalibrates the attribute channels for the original input with the weights learned.

### 4.4. Enhanced Spatial Attention

To better express the effective information of signature features, it is best to combine the channel attention mechanism SE and the improved spatial attention mechanism. The ESA mechanism used is to make features focus more on the region of interest [20]. When these highlighted features are aggregated together, we can obtain a more representative feature. When designing an attention block, several elements need to be carefully considered. First, note that the block must be light enough because it will be inserted into each remaining module of the network. As shown in Figure 7, The ESA mechanism proposed in this paper starts with the 1 × 1 convolutional layer to reduce the channel size, so that the entire block can be very lightweight. For the attention area to well describe the important information of the signature image, a large range of receptor fields is required. To expand the receiving domain, a large-step convolution is used, followed by a maximum pooling layer. In signature image classification, a combination of convolution and maximum pooling is used to quickly reduce the spatial dimension at the beginning of the network. In this paper, the max-pooling operation of large windows and striding is used, the spatial dimension of the upper sampling layer corresponding to the front part is restored, and the channel dimension of the 1 × 1 convolutional layer is restored. Finally, an attention mask is generated by an S-shaped layer, and a better way to achieve spatial attention blocks is to use non-Local blocks.

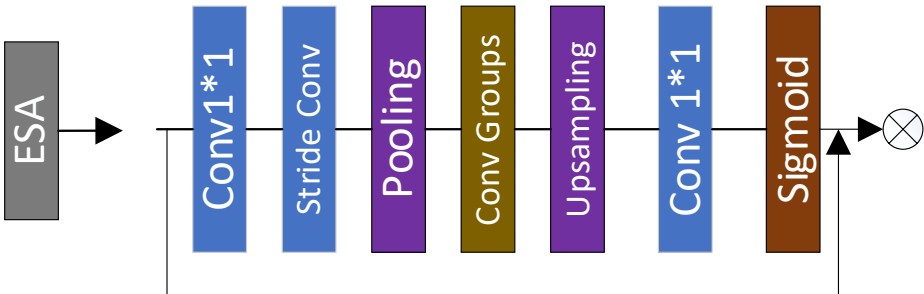

**Figure 7.** ESA module structure diagram.

ESA adaptively re-reduces the pixels of the signature image based on the spatial context, and it allows the network to learn more distinguishing features. In addition, the ESA is lighter and has better performance than ordinary space concern blocks. Figure 7 is the frame structure diagram of the ESA module:

*4.5. Loss Function*

The loss function is used to evaluate the predicted value and the real value of the model. When training the model in deep learning, the loss function is calculated and the model parameters are updated, thereby reducing the optimization error, until the loss function drops to the target value or reaches the training times [21]. This paper generates three pairs of merged features for the model by inverting the gray value of the signature image: reference signature and test signature, inverse grayscale reference signature and test signature, and reference signature and inverse grayscale test signature, as shown in Figure 4. During training, by forcing the model to make the same signature verification decision for the three pairs of merged features, the model will be directed to pay attention to the stroke information of the signature. This paper uses an inverse supervised loss function based on cross-entropy error.

Suppose $y$ is a binary ground-truth label of the test signature concerning the reference signature, where 1 means the test signature is real, 0 means forged, $\hat{y}_i(i = 1, 2, 3)$ is the reference signature and test signature, respectively, inverse grayscale reference signature and test signature, the predicted probability value of the reference signature and the inverse grayscale test signature. Based on the cross-entropy error function binary classifier, the inverse-supervised loss function of the signature image is defined as:

$$L = -\sum_{i=1}^{3} a_i [y \ln \hat{y}_i + (1 - y) \ln(1 - \hat{y}_i)] \tag{1}$$

When $y = 1$:

$$L = -\ln \hat{y} \tag{2}$$

When $y = 0$:

$$L = -\ln(1 - \hat{y}) \tag{3}$$

where are the hyperparameters that tune the three pairs of weights? $L$ is the cross-entropy loss function, when $y = 1$, the closer the predicted output is to the real sample label 1, the smaller the loss function $L$; when $y = 0$, the closer the predicted output is to the real sample label 0, the smaller the loss function $L$; whether the real sample label is 0 or 1, represents the prediction output and the gap, Unlike traditional loss functions, the inverse supervised loss function has three components.

## 5. Experimental Result

This paper tests the method proposed in this paper on two public data sets of Bengali and Hindi and two self-built data sets of Chinese and Uyghur. We also carry out experiments in the form of mixed languages.

The experimental platform of this paper is the graphics card NVIDIA 2080ti. Using the PyTorch deep learning framework and the training of the GPU acceleration network, the research on multilingual handwritten signature authentication based on an improved IDN network is realized.

*5.1. Evaluation Criteria*

This paper uses (*FRR*, *FAR*, and *ACC*) to synthesize the method proposed in this experiment to compare with other methods, where the False Rejection Rate (*FRR*) is defined as the ratio of the number of false rejections to the number of genuine signature samples, and the False Acceptance Rate (*FAR*) is only the user's false signature false judgment as to the false incidence of genuine signature [22]. Since *FRR* and *FAR* are mutually restrictive, the Total Accuracy Rate (*ACC*) is introduced for evaluation, and the higher the *ACC*, the better the ability of the proposed algorithm to handle genuine and forged signatures. The calculation formulas for the three evaluation criteria are as follows:

$$FRR = \frac{FP}{FP + TN} \tag{4}$$

$$FRR = \frac{TP}{TP + TN} \tag{5}$$

$$ACC = \frac{TP + TN}{TP + TN + FP + FN} \tag{6}$$

In the above equation, *TP* is the number of genuine signatures predicted to be real signatures, *FN* is predicted to be the number of forged signatures for real signatures, *FP* is predicted to be the number of forged signatures, and *TN* is predicted to be the number of forged signatures.

*5.2. Algorithm Implementation Process of Improved IDN Network*

The network consists of four concatenated convolution modules sharing flow weights (Algorithm 1), which can be divided into test signatures and reference signatures as input for the discriminative flow, and the reverse grayscale signature as input for the reverse flow, and then through the SE module and the ESA module. Combined with the feature transfer of discriminative flow and reverse flow, effective features are extracted.

---

**Algorithm 1** The improved inverse discriminator network

---

**1: Input:** the images reference and test, matrix size [384, 96].
**2: Initialization:** reference signature and test signature images respectively to obtain the inverse-gray reference signature and inverse-gray test signature
**3: Feature Extraction:** Conv + ReLu + Conv, Feature extraction on four input signature images.Get Ts, Tt, IGRt and IGTt of size [115, 220].
**4: SE:** For the reference, the feature extraction of the SE module is performed on the four input streams respectively.
**5: ESA:** Feature connection is made between reverse flow features and discriminative flow features.
**6: for** I = 1, 2, 3, . . . , T **do**
**7:**      **Cat:** Splicing the features of the 2 streams
**8:**      **GAP:** Perform max pooling on the spliced data to reduce the feature size
**9:**     **FC:** Mapping multi-dimensional features to a 1-dimensional vector representation
**10:**      **Classifier:** The classification result with the highest output probability
**11: end for**
**12:**      **Evaluation**: Average the three classification results and output the final discrimination result.
**13: Output**: Output is 0: Test is true; output is 1: Test is false.

---

*5.3. Chinese and Uyghur Data Set*

The Chinese data set in this paper has 100 signature samples. Each person has 20 real signature samples and 20 forged signature samples. In total, 90 of the 100 signature samples are used as training data, and the rest are used as test sets. For each person, we randomly select 2 of the 20 real signatures, a total of 190 pairs ($20 \times 19/2$). For forged signature pairs, we randomly select 1 true and 1 false, a total of 400 pairs, and randomly select 190 pairs as forged signature pairs; for reference samples and real samples, 10 real signatures are randomly selected as reference signatures and 19 forged signatures as references, forming 190 pairs of reference signatures and forged signature samples. Therefore, for each individual, we have a total of 380 pairs of samples, of which 190 are reference real pairs and 190 are reference forged pairs.

Table 1 shows the results of different methods. The comparative experiments using machine learning and deep learning, in which the identification of machine learning algorithm LTP is FRR of 9.38%, FAR of 9.00%, and ACC of 90.81%. The verification results of comparative experiments using multipath attention mechanism and reverse mechanism without improved IDN network are FRR of 8.67%, FAR of 6.50%, and ACC of 92.18%. The final discrimination rate based on the improved IDN model proposed in this paper is FRR of 5.2%, FAR of 4.46%, and ACC of 95.17%. Two sets of ablation experiments were also done in this paper, and when the IDN network and SE attention mechanism were used, the identification results of the Chinese dataset were FRR of 5.93%, FAR of 7.04%, and ORR of 93.49%. When using IDN and ESA, the identification results of the Chinese dataset were FRR of 9.51%, FAR of 5.51%, and ACC of 92.40%. The comparative experimental results show that the improved IDN model in this paper is better than other methods in the evaluation index. In the Chinese data set, the IDN model is better than other methods because it uses the attention mechanism (SE block) and the improved spatial attention mechanism (ESA). The point is that the IDN model uses the multi-path attention mechanism and reverse mechanism to extract the important features of the signature image more effectively, which proves the effectiveness of the SE-block and ESA mechanism used in this paper.

**Table 1.** Signature verification results and comparative experiments of Chinese dataset.

| Authors | Data/(Person) | Method | Type | FRR% | FAR% | ACC% |
|---|---|---|---|---|---|---|
| Liu, L. [9] | Chinese (1243) | SigNet | WI | - | - | 90.09 |
| Liu, J. [20] | Chinese (2880) | CNN-OSV | WD | 11.52 | 10.51 | 82.75 |
| Masoudnia, S. [13] | Chinese (249) | AlexNet | SVM | 7.50 | 5.00 | 87.50 |
| Wei, P. [16] | Chinese (749) | IDN | WI | 5.47 | 11.52 | 90.17 |
| Ours | Chinese (100) | LTP | SVM | 9.38 | 9.00 | 90.81 |
| Ours | Chinese (100) | IDN | WI | 8.67 | 6.50 | 92.18 |
| Ours | Chinese (100) | IDN + SE | WI | 5.93 | 7.04 | 93.49 |
| Ours | Chinese (100) | IDN + ESA | WI | 9.51 | 5.51 | 92.40 |
| Ours | Chinese (100) | Our method | WI | 5.2 | 4.46 | 95.17 |

The Uyghur data set used in this paper also has 100 signature samples. As with the Chinese data set, each person has 20 real signature samples and 20 forged signature samples. A total of 90 of the 100 signature samples are used as training data, and the rest are used as test sets. For each person, we randomly select 2 of the 20 real signatures, a total of 190 pairs ($20 \times 19/2$), for forged signature pairs, we randomly select 1 true and 1 false, a total of 400 pairs, randomly select 190 pairs as reference samples and real samples of forged signature pairs, randomly select 10 real signatures as reference signatures and 19 forged signatures as references, forming 190 pairs of reference signatures and forged signature samples. Therefore, for each individual, we have a total of 380 pairs of samples, of which 190 are reference real pairs and 190 are reference forged pairs.

From Table 2, the most effective features of the signature image are extracted from the experiments based on the IDN model and the SE-block mechanism, and the improved spa-

tial mechanism proposed in this paper, and then compared with other algorithms through the Gap global average pooling layer; it can be seen from Table 2 that the discrimination rate of the self-built Uyghur dataset in the IDN network is an FRR of 6.05%, FAR of 8.05%, ACC of 92.96%. When using the ablation experiments done on the Uyghur dataset, the best identification rate was 6.84% for FAR, 6.50% for FRR, and 93.56% for ACC. The verification rate of the improved IDN method used in this paper is an FRR of 7.47%, FAR of 3.89%, and ACC of 94.32%, and the total correct rate is 1.36% higher than that without the improved network. The algorithm proposes the most effective features for the Chinese and Uyghur samples constructed in this paper.

**Table 2.** Results of Uyghur data sets and comparative experiments.

| Authors | Data/(Person) | Method | Type | FRR% | FAR% | ACC% |
|---|---|---|---|---|---|---|
| Aini, Z [23] | Uyghur (30) | GLCM | BP | 5.30 | 7.00 | 91.06 |
| Ghaniheni, Z [24] | Uyghur (30) | Directional feature | KNN | 9.09 | 5.75 | 92.58 |
| Ours | Uyghur (100) | LTP | SVM | 18.29 | 12.33 | 84.37 |
| Ours | Uyghur (100) | IDN | WI | 6.05 | 8.05 | 92.96 |
| Ours | Uyghur (100) | IDN + SE | WI | 6.84 | 6.50 | 93.56 |
| Ours | Uyghur (100) | IDN + ESA | WI | 6.10 | 9.21 | 92.35 |
| Ours | Uyghur (100) | Our method | WI | 7.47 | 3.89 | 94.32 |

*5.4. BHSIG260 Open Data Set*

The public data set BHSIGH260 used in this paper contains two data sets, namely BHsig-B (Bengali) and BHsig-H (Hindi). Among them, BHsig-B contains 100 personal signatures, and each signature contains 24 genuine signatures and 30 forged signatures. When using BHsig-B, in the data set of 100 people, we use the sample of 90 people as the training set and the remaining 10 people as the test set. For everyone, we have $276 = (24 \times 23/2)$ pairs of reference and real samples. Another public data set BHsig-H contains 160 signatures, and each signature contains 24 genuine signatures and 30 forged signatures. In the experiment, we selected 130 signatures from 160 signatures as the training set and the remaining 30 as the test set. Similarly, for each person, we have $276 = (24 \times 23/2)$ pairs of reference and real samples. Each person's signature sample randomly selects 10 real signatures as reference signatures and 19 forged signatures as reference signatures, forming 190 pairs of reference signatures and forged signature samples. So, there are 380 samples for each person, of which 190 pairs are reference real pairs and 190 pairs are reference fake pairs.

The results show that the improved IDN model in this paper is compared with the IDN model. Table 3 shows the experimental results of two public data sets. In the table, WI represents the method of independent writers to build the same model for any writer, and WD represents the method of independent writers to train different models for each writer, which usually requires more samples for training. It is worth noting that the author-dependent method adopts different independent methods of training. Here, we list the author-dependent methods as references. On these data sets, the improved IDN model in this paper is superior to other methods in evaluation criteria, which proves the effectiveness of this method.

**Table 3.** Results of BHSIG260 data sets and comparative experiments.

| Authors | Dataset | Method | Type | FRR% | FAR% | ACC% |
| --- | --- | --- | --- | --- | --- | --- |
| Dey, S [25] | BHsig-B | SigNet | WI | 13.89 | 13.89 | 86.11 |
| Dey, S [25] | Bhsig-H | SigNet | WI | 15.36 | 15.36 | 84.64 |
| Dutta, A [26] | Bhsig-B | LBP, ULBP | WD | - | - | 66.18 |
| Dutta, A [26] | Bhsig-H | LBP, ULBP | WD | - | - | 75.53 |
| Ours | Bhsig-B | IDN | WI | 5.42 | 4.12 | 95.32 |
| Ours | Bhsig-H | IDN | WI | 4.93 | 8.99 | 93.04 |
| Ours | Bhsig-B | IDN + SE | WI | 3.67 | 1.61 | 96.71 |
| Ours | Bhsig-H | IDN + SE | WI | 11.87 | 2.44 | 95.77 |
| Ours | Bhsig-B | IDN + ESA | WI | 3.79 | 11.78 | 96.70 |
| Ours | Bhsig-H | IDN + ESA | WI | 2.72 | 4.68 | 95.68 |
| Ours | Bhsig-B | Our method | WI | 3.14 | 1.50 | 97.17 |
| Ours | Bhsig-H | Our method | WI | 6.65 | 2.31 | 96.86 |

*5.5. Experimental Result of Mixing Two Languages*

This chapter adopts the experiment of mixing two languages. Its data set is composed of five data sets including Chinese, Uyghur, Bengali, and Hindi. Each person in the self-built data set contains 20 true signatures and 20 forged signatures. The public data set contains 24 true signatures and 30 forged signature samples. When every two languages are mixed, the number of training sets and test sets is selected according to the number of mixed languages.

In the mixed languages, 180 signatures of Chinese and Uyghur are selected as the training set, the rest as the test set, while 160 signatures of mixed Chinese and BHsig-B are selected as the training set, and the rest as the test set. Among the 260 true and false signatures of Uyghur and BHsig-H, 220 were used as the training set and 40 as the test set. In total, 160 of the 200 true and false signatures of Uyghur and BHsig-B are selected as the training set, the rest as the test set, 220 in Chinese and BHsig-H are selected as the training set and 40 as the test set. In this mixed language experiment, for each person, we randomly select 2 of 20 real signatures, a total of 190 pairs ($20 \times 19/2$); for forged signature pairs, we randomly select 1 true and 1 false, a total of 400 pairs, and randomly select 190 pairs as forged signature pairs. Therefore, for each individual, we have a total of 380 pairs of samples, of which 190 are reference real pairs and 190 are reference forged pairs.

The results show that the improved IDN model in this paper is better than other methods in evaluation indicators. On the public dataset, the attention mechanism SE and the improved spatial attention mechanism are used, which is better than the multi-attention mechanism SE and ESA. The mechanism extracts the important features of the signature image more efficiently. This experiment also uses the WI model, which mixes two different languages to build the same model for any author, improving the usefulness of mixed languages. It can be seen from the Table 4 that the model or feature extraction algorithm proposed in this paper is very effective. The identification results of the mixture of Chinese and Uyghur languages achieve an FRR of 10.50%, FAR of 2.06%, and ACC of 96.33%.

Thus, it not only improves the identification results of public datasets but also improves the identification rate of the datasets built in this paper.

**Table 4.** Signature verification results and comparative experiments of open data sets.

| Dataset | Method | Type | FRR% | FAR% | ACC% |
|---|---|---|---|---|---|
| Uyghur + Chinese | IDN | WI | 9.16 | 7.87 | 91.87 |
| Chinese + BHsig-B | IDN | WI | 15.36 | 15.36 | 84.64 |
| Chinese + BHsig-H | IDN + SE | WI | 7.10 | 14.46 | 89.22 |
| Uyghur + BHsig-B | IDN + SE | WI | 18.55 | 21.15 | 88.49 |
| Uyghur + Chinese | IDN + SE | WI | 5.08 | 4.47 | 95.40 |
| Han + BHsig-H | IDN + ESA | WI | 15.78 | 31.5 | 83.45 |
| Uyghur + Chinese | IDN + ESA | WI | 7.92 | 8.28 | 91.89 |
| Uyghur + BHsig-H | IDN + ESA | WI | 2.29 | 28.73 | 84.48 |
| Uyghur + Chinese | Our method | WI | 10.5 | 2.06 | 96.33 |
| Chinese + BHsig-B | Our method | WI | 7.30 | 8.07 | 92.33 |
| Chinese + BHsig-H | Our method | WI | 3.56 | 10.15 | 93.19 |
| Uyghur + BHsig-B | Our method | WI | 3.21 | 8.78 | 94.00 |
| Uyghur + BHsig-H | Our method | WI | 7.47 | 12.36 | 90.07 |

*5.6. Comparison of Experimental Results*

Signature is closely dependent on language. Individuals using different languages have different writing habits and styles. This paper always uses signature samples in four different languages.

Table 5 shows that the results of the single language data set are very good. Although the experimental results of the mixing of the two languages are relatively lower than those of the single language, the combination of several effective network frameworks such as IDN network, cascaded convolution module, attention mechanism SE module, and improved spatial mechanism ESA under the WI mode proposed in this paper obtains the best signature authentication effect. Especially in multi-ethnic and multilingual areas such Xinjiang, it brings convenience to many fields. Table 5 shows the experimental comparison of the signature authentication methods used in this paper:

**Table 5.** Comparison of other research results.

| Authors | Method | Type | Database | FRR% | FAR% | ACC% |
|---|---|---|---|---|---|---|
| Chattopadhyay, S [27] | ReSent + 2D attention | WI | BHsig-H | 8.98 | 12.01 | 89.50 |
| Ours | Our method | WI | BHsig-H | 3.75 | 2.57 | 97.20 |
| Manna, S [28] | self-supervised learning (SSL) | WI | Chinese | 58.30 | 27.80 | 64.68 |
| Ours | Our method | WI | Chinese | 5.2 | 4.46 | 95.17 |
| Zhang, S [14] | Bovw | WD | Uyghur | 3.58 | 8.81 | 93.81 |
| Ours | Our method | WI | Uyghur | 7.47 | 3.89 | 94.32 |
| Hamadene, A [2] | CT, DCCM | - | CEDAR + GPDS | 16.32 | 16.80 | - |
| Ours | Our method | WI | Chinese + Uyghur | 10.5 | 2.06 | 96.33 |

**6. Conclusions**

This paper takes multi-language offline handwritten signature images as the research object and proposes an improved IDN network. Since there is currently no suitable multilingual offline handwritten signature data set, this paper first collects a challenging multilingual data set. During the experiment, the collected signature images are normalized and binarized and then used for the author's independent handwritten signature verification. The IDN network contains four shared streams of weights; where two discriminative streams extract the features of the signature image, and two reverse streams focus on the signature features. The channel attention mechanism SE and the improved spatial attention mechanism ESA are utilized to solve the stroke sparsity problem in signature verification and suppress useless feature information. In the testing process, the model adopted in this paper inputs a reference signature image and a side signature image, respectively, and outputs the true or false test signature. From the above series of experiments, it can be proved that this method not only effectively improves the public data set but also effectively

identifies the self-built data set, whether it is a signature of a single language or a mixture of two languages. For future work, more languages will be collected, such as Kazakh, Kirgiz, and other minority languages, etc., and more signatures of mixed languages will be used for signature verification and a recognition combined system.

**Author Contributions:** N.X. conceived, designed, N.X. and Z.Y. performed the experiments; N.X. contributed materials; N.X. and A.A. conducted the formal analysis. N.X. wrote, reviewed and edited the paper, M., A.A, Z.Y.; K.U. revised the manuscript and supervised. All authors have read and agreed to the published version of the manuscript.

**Funding:** In This work was supported by the National Natural Science Foundation of China (No. 61862061, 61563052, 61163028), Scientific Research Initiate Program of Doctors of Xinjiang University under Grant No. BS180268, and the Funds for Creative Groups of Higher Educational Research Plan in Xinjiang Uyghur Autonomous, China (No. XJEDU2017TO02).

**Institutional Review Board Statement:** Not applicable.

**Informed Consent Statement:** Not applicable.

**Data Availability Statement:** The self-built data sets used in this paper, Chinese and Uyghur data sets, have not been made public. This is the data set collected by our college, which will be published soon.

**Acknowledgments:** The authors are very thankful to the editor and the referees for their valuable comments and suggestions for improving the paper.

**Conflicts of Interest:** The authors declare no conflict of interest.

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
