# Peer review of "Multilingual Offline Signature Verification Based on Improved Inverse Discriminator Network"

_information, doi:10.3390/info13060293_

Round 1

Reviewer 1 Report

Review of article: information-1694187

“Multilingual offline signature verification based on improved inverse discriminator network”

The authors present an improved verification network to determine the true or false signature for an offline signature verification system using an attention mechanism.

After reading the article I have some comments about it. The idea of the use of an IDN seems to be a good one together with the SE and ESA modules. Nonetheless, because of the writing style, it is very difficult to follow the kern ideas of the article. Also, the numbers expressed in the result tables seem to be very promising. To improve the article I suggest carrying on an ablation study to determine the importance of the different parts of the proposed system. Also please explain how you realize the training of the network, the number of training, test, and verification images, and how the Loss function behaves in this case, etc. Also, comment if you have implemented the other algorithms that are used as a comparison or if you have taken their published results.

Please improve your writing style, in the current state, it is very difficult to read it as I have already mentioned. Review your equations please and correct the equations (2) and (3), also define all the variables that you use in all the equations. In the abstract also it is better if you say something about the numerical results and the goodness of your approach.

Improve the use of the bibliographic citation, the use of the comma sign (,) and the et al. expression. The 4.2 and 4.3 points have de same description: “Overall structure of the paper”

Here are some of the writing errors as an example of what you need to improve:

of inverse grayscale image;  image are the verification stream; the proposed the method requires

life.4. Proposed 147 Method. ; feature channels is C- 238 2 is….

Author Response

Dear reviewer:

Thanks very much for taking the time to review this manuscript. We really appreciate all your generous comments and suggestions!

We have uploaded your revised comments as shown in the attachment:

Reviewer 2 Report

The article is interesting, original, contains new results, and can be published.

Author Response

Dear reviewer:

Thanks very much for taking the time to review this manuscript. We really appreciate all your generous comments and suggestions!

Reviewer 3 Report

High quality paper with outstanding contribution which can serve urgent publication. The focus of the paper is worthy of investigation and significantly advance our knowledge.

Author Response

(The authors gave the same response as above.)

Reviewer 4 Report

This paper presents an off-line handwritten signature verification of monolingual and multilingual mixture and proposes an improved verification network (IDN), which adopts user-independent  handwritten signature verification, to determine the true signature or false signature.

The manuscript is quite well written and the experimental results and comparisons reported are interesting.

However I have some comments to do:

  • the Authors must add a pseudocode of the overall framework.
  • the evaluation criteria and the abbreviations must be better explained; furthermore using confusion matrices can better show the performance.
  • in the Introduction, citations must be formatted correctly.

Author Response

(The authors gave the same response as above.)

Reviewer 5 Report

The paper presents a method for improving the accuracy of off-line signature verification methods. It not only improves the signature verification method, but also builds a handwritten multilingual database with 2 different types of characters, which is a rather rare possibility for the community to study.

The authors have a rather long history in the challenging field of the offline signature recognition in the Uyghur language, and also recently extended to English and simplified Chinese, which is proven by reference 26 and the references therein.

The proposed method is a combination of four neural networks applied on the original, preprocessed and the inverse grayscale image of the signatures. The method seems plausible and carefully built, it improves the accuracy of other methods, though the False reflection rate is not better for Uyghur, for the other languages all the used metrics improve. I think, the used reference methods are sufficiently up-to date.

The signature sample database will be also very helpful the researchers, as the simplified Chinese and the Uyghur characters represent completely different writing concept, in contrast with the Bengali and Devanagari, in the other used database. However, it is not really clear, whether the forged signatures are trying to mimic the original person's signature whether they have seen the original signatures, or the people were just writing the name in their own handwriting. It would be advisable to clarify the conditions a little bit more.

The text is very nicely written, it is understandable. The structure is rather logical, though in some cases the description of some building blocks (together with the resolution of the shorthand notation) comes much after the first usage of the block, and some parts are not perfectly clearly described.

The figures help the understanding of the concept and the results, though they are often not too aesthetic as the texts that ought to represent parts of the flowchart are neither aligned nor of the same size, moreover, in many cases it is shrunk to one direction forming thin or flat letters.

I suggest to consider the following comments to improve the paper

1.) The names of the authors and their e-mail addresses and the abbreviation of their names in the affiliation under the main title and in the side panel are not consequent, please correct them.

2.) The text in line 41 on page 1 sounds weird: Many countries and China..
Many countries including China might be better.

3.) Maybe the short section about the metrics for performance (subsection 5.1 around Equations (4), (5) and (6) ) would be better in Section 2. In Section 2 the results are written very compactly, this might help the comparison of the results.
Or at least at the first mention of FAR, FRR, AER, etc. it would be helpful to write the full name not only the abbreviation (similar holds for other abbreviations, too, though GLCM is resolved at least twice in Section 2, which is unnecessary).

4.) There are some randomly placed numbers in the texts, that I suppose are citations without [ ] around them (or some citations with [ ] around them).

5.) Abdul-Haleem's reference [3] is referenced as 2

6.) Ajij et al. has the reference number 5 between the name of the first author and the "et al."

7.) Figure 1 has not uniform texts in the boxes, one is capitalized, the other i full small letters

8.) The caption of figure 2 is not  clear, it could be understood as if the left hand side 2 signature examples are the original ones and the right hand side 2 are the forged ones (for all 4 writings), or as if one set on the right of the dashed line contains a true and a forged signature. Please be a bit more precise with the caption.

9.) Figure 3 has not aligned texts and weirdly stretched letter types, as if it was compressed only in the vertical direction.

10.) The title of Subsection 4.2 is misleading,it is not the structure of the paper, but hte structure of the method

11.) ESA, SE, are used before writing their full names on page 5

12.) Figure 4 has not proportionally shrunk letters, they are thin (compared to fig 3 where they are flat), moreover the teal arrow at the right hand side does not start from the teal (cyan) aggregator.
Maybe separating the SE-ESA part into a block and then expanding it in a separate plot would be also possible.
Also the conv-ReLU-conv part is not properly connected it looks as if all the inputs would be merged, and then the separation afterwards is not given visually

13.) it would be great to use the same colours to the pairings in Figures 3 and 4.

14.) Figures 5 and 6 are also expanded upwards, whereas Figure 7 in horizontal direction

15.) I don't see the references to 19-27 inside the text, only in the tables, but there they are written together with the method name. It would be advisable to separate citation form the table texts, it is very hard to find and identify them.

16.) In the  Bibliography, reference 17 has an extra space in the front, and the use of "et al." is not consequently used in the references.

Author Response

(The authors gave the same response as above.)

Round 2

Reviewer 1 Report

Review of article: information-1694187

"Multilingual offline signature verification based on improved inverse discriminator network"

 Second Review of the paper. The authors have already submitted to review a version of this article. After a quick reading, I can notice an improvement in this version. Nonetheless, the main drawbacks of the first version remain. For example, in equation number one, what means the variable ai? Also, notice that equations 4 and 5 are the same, so you have not defined the variable FAR.

 Also, several sentences don't have a clear meaning. I list some of them below:

 -          two of which accept the feature of the original signature image are the verification stream

 -          The ESA of the attention module connects discriminant flow and reverse flow to realize message propagation.

 -          The network comprises 4 weight-sharing streams consisting of 2 streams are the discriminative streams

 -          Using the algorithm proposed in this paper, improve the identification rate of of-fline handwritten signatures

 -          Suppose is the truth label of the test signature relative to the reference signature,

 Also, I made some suggestions in the first revision that, in my opinion, should be taken into account: "To improve the article, I suggest carrying on an ablation study to determine the importance of the different parts of the proposed system. Also, please explain how you realize the network's training, the number of training, test, and verification images, how the Loss function behaves in this case, etc."

 Consequently, I recommend that the authors rework the article considering the above comments. In another case, please comment on why it is not necessary to carry on the mentioned suggestions. 

Author Response

Thank you very much for your support and comments on our paper. We have modified your comments, as shown in the attachment.

Reviewer 4 Report

The Authors have refined this work during this review cycle and have substantially satisfied all of my comments and suggestions.

This manuscript can be accepted in its current form now.

Author Response

(The authors gave the same response as above.)

Round 3

Reviewer 1 Report

Review of article: information-1694187

"Multilingual offline signature verification based on improved inverse discriminator network"

The authors present a third review of the paper. The authors have already submitted the first and second versions of this article to review. There are some advances in the article's writing, especially if you consider the comments placed in the authors' letter. Nonetheless, the main drawbacks of the first and second versions remain.

- The equations are not well written

- There are still several sentences that are very hard to understand (I have mentioned in the second review only a few of the total)

In my opinion, the arguments about the no existence of an ablation analysis are not very convincing.

So, in consequence, I still recommend the authors rework the article. Take care of what you write in the text and submit it again.